# Novel Interventions on Comorbidities in Patients with Fetal Alcohol Spectrum Disorder (FASD): An Integrative Review

**DOI:** 10.3390/biomedicines12030496

**Published:** 2024-02-22

**Authors:** Vicente Andreu-Fernández, Nunzia La Maida, Maribel Marquina, Afrouz Mirahi, Oscar García-Algar, Simona Pichini, Adele Minutillo

**Affiliations:** 1Grup de Recerca Infancia i Entorn, Institut d’Investigacions Biomèdiques August Pi i Sunyer (IDIBAPS), 08036 Barcelona, Spain; viandreu@recerca.clinic.cat (V.A.-F.); ogarciaa@clinic.cat (O.G.-A.); 2Biosanitary Research Institute, Valencian International University, 46002 Valencia, Spain; mariaisabel.marquina@universidadviu.com; 3National Centre on Addiction and Doping, Istituto Superiore di Sanità, 00161 Rome, Italy; nunzia.lamaida@iss.it (N.L.M.); simona.pichini@iss.it (S.P.); 4Department of Neonatology, ICGON, Hospital Clínic-Maternitat, BCNatal, 08028 Barcelona, Spain; afrouz.mirahi@gmail.com; 5Department of Surgery and Medical-Chirurgical Specialties, University of Barcelona, 08007 Barcelona, Spain

**Keywords:** prenatal exposure delayed effects, fetal alcohol spectrum disorders, interventions, early diagnosis, follow-up studies

## Abstract

Prenatal exposure to alcohol can cause Fetal Alcohol Spectrum Disorders (FASDs) after birth, encompassing a spectrum of physical, cognitive, and behavioral abnormalities. FASD represents a severe non-genetic disability avoidable through alcohol abstinence during pregnancy and when planning it. Clinical severity depends on alcohol impact, symptomatology, and resulting disabilities. FASD is a permanent disability with no recognized specific medical care. Conversely, secondary FASD-related disabilities can be symptomatically treated. This integrative review aims to provide information about the novel pharmacological treatments of FASD-associated comorbidities by selecting the last ten years of studies carried out on animals and humans. PRISMA guidelines were followed to search human/animal model studies of pharmacological interventions on FASD comorbidities, using different databases (PubMed, Cochrane, etc.). From 1348 articles, 44 met the criteria after full-text analysis. Firstly, all the reported studies point out that early diagnosis and tailored interventions are the principal tools to reduce FASD-related secondary disabilities, due to the fact that there is currently no approved pharmacological treatment for the tissue damage which produces FASD. Despite limitations in study designs and small sample sizes, these review results highlight how the treatment strategies of children with FASD have changed. In the past, studies focused on treating symptoms, but in the last years, researchers have turned their attention to the prevention targeting central nervous system embryogenesis. Novel treatments like choline and natural antioxidants and nutritional supplements are the most investigated treatments in humans with promising results. More follow-up studies need to be performed, to confirm and generalize reported efficacy to a wide sample size.

## 1. Introduction

Prenatal Exposure to Alcohol (PAE) can exert toxic effects on the fetus, causing a broad range of physical, cognitive, and behavioral abnormalities that are classified as Fetal Alcohol Spectrum Disorders (FASDs) [1,2,3]. FASD is the primary non-genetic cause of a severe permanent disability, which can be totally prevented by abstaining from alcohol consumption during pregnancy. FASD is a complex set of clinical conditions that vary depending on the damage caused by alcohol exposure, the severity of symptoms observed, and the type of disability exhibited [4] with Fetal Alcohol Syndrome (FAS) being the most severe condition.

Growth deficits, craniofacial dysmorphology, birth malformations, and Neurodevelopmental Disorders (NDs) are clinical and phenotypic manifestations of FASD that can vary in severity and expression according to dose, amount, and frequency of consumption, as well as the gestational period of exposure, maternal nutrition, and genetic background [5,6,7,8]. FAS, partial FAS (pFAS), Alcohol-Related Neurodevelopmental Disorder (ARND), and Alcohol-Related Brain Disorders (ARBDs) have specific diagnostic criteria based on the co-occurrence of one or more clinical and above-mentioned phenotypic manifestations. However, a specific neuropsychological profile has not yet been established for patients affected by FASD [9].

The cognitive and behavioral difficulties are a consequence of Central Nervous System (CNS) permanent damage due to PAE [10,11]. Some of the difficulties commonly referred to as Primary Disabilities (PDs) include intellectual disability, deficits in executive functioning, memory and information processing, language delay or dysfunction, attention deficits and hyperactivity, and general difficulty in learning [12,13,14,15]. If these are not promptly treated, they can lead to Secondary Disabilities (SDs), that develop later in life and include health problems, employment issues, failed school experiences, legal issues, inappropriate sexual behavior, drug and/or alcohol use problems.

Early diagnosis allows for the correlation of symptoms with an accurate diagnosis and treatment and helps to minimize SD. Indeed, SDs decrease significantly in adolescence and in adulthood if the intervention starts before the age of six, due to the large neuronal plasticity of a child’s brain [16,17].

Moreover, early diagnosis cannot be carried out in cases of abandoned children when maternal alcohol consumption is unknown or when they rarely admit using ethanol during pregnancy [18].

Late diagnosis in adulthood leads to greater treatment challenges, as the pathological mechanisms have already been established. However, it is still possible to activate pathways of awareness, treatment, and support by starting from the family situation and the emerged disabilities [19].

Language problems, which are characterized as difficulties in expression and understanding, require special attention in individuals with FASD [20,21]. In FASD individuals, metacognition is reduced or absent and it is associated with language impairment, preventing them from expressing their emotions verbally. Emotions are often the consequence of a painful experience resulting in physical discomfort (e.g., tachycardia, abdominal pain, etc.) [22]. Moreover, FASD individuals are not able to distinguish their own internal reality from the external one, resulting in their inability to grasp the difference between dreams, fantasies, beliefs, and hypotheses. Thinking, reflecting, and reasoning about one’s own and others’ mental states is another challenge for FASD individuals [23].

FASD is related to five comorbid conditions with the highest aggregate prevalence (between 50% and 91%) including abnormal findings on functional studies of the peripheral nervous system and special senses, conduct disorder, receptive language disorder, chronic serous otitis media, and expressive language disorder [24].

Moreover, the most common comorbidities concern the diagnoses of Cognitive Delay, Attention Deficit/Hyperactivity Disorder (ADHD), Specific Learning Disorders (SLDs), Pervasive Developmental Disorders (autism and other pervasive disorders), and Externalizing Disorders (Oppositional Defiant Disorder, Conduct Disorder, Intermittent Explosive Disorder) [25,26]. Neurobehavioral Disorders associated with Prenatal Exposure to Alcohol (NDs-PAE) have been well defined in the Diagnostic and Statistical Manual of Mental Disorders [27], and represent a specific diagnostic category highlighting the evolution of the disorder throughout the life span.

The treatment of FASD requires an integrated approach, involving different disciplines (psychiatry, neurology cardiology, nutritional rehabilitation courses, etc.) and professionals [28,29]. To date, there are no specific treatments for FASD. Hence, it is necessary to evaluate each affected subject individually to determine an appropriate targeted treatment and improve the quality of life [30,31].

Due to their neurocognitive particularities, individuals with FASD may not be consistently able to adhere to a psychotherapeutic process and complete it successfully. Cognitive-Behavioral Therapy (CBT) and Parent Training are successful therapeutic interventions supporting both FASD patients and the whole family group [32].

Sometimes, despite optimization of behavioral interventions, some individuals will need pharmacological treatments for a long period to manage their symptoms and comorbidities [33].

Drugs commonly used in the FASD clinical manifestations are different for clusters of signs and symptoms. Adrenergic agents are the primary treatment for hyperactivity including irritability, aggression, insomnia, agitation, and anxiety. Mood stabilizers (such as Divalproex and Lamotrigine) can be administered to treat conditions associated with emotional regulation (e.g., anxiety, depression, mood swings, etc.), while SSRIs (Fluoxetine, Citalopram, Sertraline) are used as a second option for both these symptom groups, but they are not recommended for preschool children. Concerning disorders linked to executive functions and/or hyperactivity (e.g., impulsivity, poor concentration, restless movement), stimulant-based amphetamines (like Lisdexamfetamine and Dexedrine) are recommended as the first-line treatment, followed by other stimulants such as Methylphenidate, Atomoxetine, or Bupropion. Finally, for symptoms pertaining to cognitive inflexibility, such as perspective disorders, frustration tolerance, social skills, reasoning, etc., atypical neuroleptics like Risperidone are recommended as the first-line treatment, while Olanzapine and Aripiprazole are suggested as second-line options.

The aim of this integrative review is the analysis of the pharmacological interventions for the treatment of FASD and its associated comorbidities such as anxiety, depression, hyperactivity, and Attention Deficit Disorder (ADHD), aggressiveness, etc. Moreover, this review analyzes novel treatments for FASD or its comorbidities in either preclinical and analytical studies or clinical trials, with special mention to treatments with natural antioxidants and choline during the perinatal period and infancy. We could not consider a meta-analysis due to the differences in the experimental design depending on preclinical studies in FASD-like animal models or trials in FASD subjects. In addition, the differences of the molecular mechanism targeted by the diverse pharmacological interventions evaluated in this review would generate an important bias in the statistical results in a meta-analysis.

## 2. Materials and Methods

Preferred Reporting Items for Systematic Reviews and Meta-Analysis (PRISMA) statement was the methodology selected for the present integrative review [34,35]. According to guidelines of the 2009 as well as the update of the 2020 PRISMA statement [36], the research team evaluated the following items: definition of the research question, hypothesis and objectives; bibliographic search; data collection, evaluation, synthesis, and comparison; critical evaluation of the scientific papers selected and finally, analysis of the main findings and conclusions including the strengths and weakness of these studies (Figure 1).

PubMed (MeSH), Cochrane Central Register of Controlled Trials, Embase, ICTRP, and CINAHL were the electronic databases consulted to collect the data. We performed an initial search using the following descriptors (as MeSH terms or not) and similar terms with the Boolean operators (AND/OR) in multiple combinations “((fetal alcohol spectrum disorders) OR (fetal alcohol syndrome)) AND ((pharmacological treatment) or (pharmacological intervention)) or (pharmacology)” “((fetal alcohol spectrum disorders) OR (fetal alcohol syndrome)) AND (choline)” “((fetal alcohol spectrum disorders) OR (fetal alcohol syndrome)) AND (natural antioxidants)” to focus the scope of this review on those pharmacological interventions whose effects on FASD or associated comorbidities have been analyzed in both humans and animal models studies. MeSH terms also included various expressions of FASD and psychotropic or antipsychotics medications as anxiolytics. All terms were adapted for the databases consulted for this review.

Inclusion criteria comprised papers written in English (with no geographical restrictions) published from 26 June 2013 to 26 June 2023 with evidence of FASD diagnosis and pharmacological treatment. Papers before 2013 were not included because they mainly focused on the PAE effects investigation rather than research based on novel treatments for FASD comorbidities exhibited both in animal models and humans.

The last ten years were selected to include the following: current information regarding novel pharmacological treatments or nutritional supplements applied to individuals with FASD; the presence of the selected terms in the title or as keywords; original research performed in humans or animal models when focused on the use of therapeutic drugs or novel pharmacological therapies to treat FASD or a specific comorbidity associated with FASD.

The types of selected experimental designs were clinical studies, clinical trials, comparative studies, case-control studies, longitudinal cohorts, cross-sectional and case report studies with a sample size of a minimum of 10 participants, as well as preclinical studies with animal models. Selected outcomes were evidence of neuropsychological or behavior evaluation as well as the measure of specific biomarkers of CNS in human or animal models. Exclusion criteria were non-systematic reviews, lack of a control or comparative group with no pharmacological treatment, experimental designs using different drugs to treat different symptoms interfering with the effects of a specific treatment, the presence of genetic or other diseases in FASD subjects which could interfere with the alterations analyzed in this review, and papers in which output variables were not related to pharmacological treatments for FASD.

The researchers M.M., V.A.F., A.M., N.L.M. and S.P. performed the initial selection of original manuscripts by screening titles and abstracts, creating a reference list of papers for the topics evaluated in the present review using Rayyan software version 1.3.3 to perform systematic reviews [37]. Four investigators (M.M., V.A.F., A.M. and S.P.) conducted all of the stages of the studies’ selection, deleted duplicate inputs, and reviewed studies as excluded or requiring further assessment. All data were extracted by two investigators (M.M. and V.A.F.) and cross-checked by the other four investigators (N.L.M., A.M., S.P., and O.G). In the case of discrepancies in the selected studies, we opted for reconciliation through team discussion. Moreover, the reference lists of some selected papers were manually checked to include some high impact additional studies from the bibliography of the original papers or meta-analyses which were relevant to the topics addressed following the snowball strategy (Figure 1). The information evaluated from each study included the following: pharmacological intervention, first author, year of publication, objective, experimental design (treatment duration and study groups), and number of participants in treated and control group; main outcomes/findings; conclusions; strengths and limitations (including biases). The eligibility criteria followed the PICOS approach (patient, intervention, comparators, outcome, and study design); population included infants diagnosed with FASD or FASD-like animal models; intervention included any dose of pharmacological treatment selected for this review; comparators included placebo, if applicable; the primary outcome was the neurocognitive or behavioral response of the FASD individuals after a pharmacological treatment; changes in levels or expression of selected molecular biomarkers related to CNS alterations and diverse neurological functions. All authors performed a critical appraisal for the selected studies following the inclusion criteria, as well as analyzed the methodology and key results.

The outputs evaluated following PRISMA were heterogeneous by the need to include animal models as well as human studies; the different populations of analyzed infants; the limited sample size observed in many of these studies and the few randomized trials using novel molecules as a pharmacological treatment for FASD or its associated comorbidities. The studies identified through database searching, selection after meeting the inclusion criteria, and the application of the exclusion criteria are indicated in Figure 1. All citations were collated and imported into Mendeley citation manager. This systematic review protocol was registered in PROSPERO (International Prospective Register of Systematic Reviews, on 27 June 2023, with the registration number CRD42023439958, available from: https://www.crd.york.ac.uk/prospero/display_record.php?RecordID=439958 (accessed on 12 January 2024).

## 3. Results

### 3.1. Characteristics of the Included Studies

A total of 1348 published articles were identified after bibliographic search in the databases indicated in the Materials and Methods Section. A total of 23 studies were included via citation searching following the snowball strategy. Once duplicate papers were deleted (893), the abstract and title of the potentially relevant 478 articles were reviewed. After applying inclusion and exclusion criteria, 330 references were eliminated. The full text of the remaining 148 references was carefully analyzed, removing 104 articles because they did not meet eligibility criteria. Since they were not related to the topic or the analyzed outcomes, they were in vitro studies or they referred to a non-pharmacological treatment. Finally, eligible and included studies in this integrative review are shown in the flowchart of Figure 1.

### 3.2. Results from Animal Studies

Animal studies explored the effects of various interventions on alcohol exposure and its neurological consequences, although not all studies were conducted considering the condition of prenatal alcohol exposure (Table 1).

These interventions included choline, minocycline, cannabidiol, omega-3 fatty acids, metformin, Shh signaling activation, rapamycin, DAT inhibitor CE-123, curcumin, crocin, Dihydromyricetin (DHM), Epigallocatechin Gallate (EGCG), folic acid, betaine, osmotin, adiponectin, Trichostatin A (TSA), Docosahexaenoic acid (DHA), papaverine, and HX106.

The findings revealed various promising outcomes. Choline demonstrated efficacy in preventing developmental anomalies, motor difficulties, and cognitive impairments linked with PAE [38,39,40,41,42]. Omega-3 fatty acids were found to reverse deficits associated with PAE, specifically in male animals [43]. Metformin exhibited significant effects in reducing oxidative stress and apoptosis in the hippocampus [44]. Some interventions, such as Rapamycin, Curcumin, DHM, EGCG, and TSA, showed potential in reversing ethanol-induced deficits in memory, neuronal damage, or neurobehavioral alterations [45,46,47,48,49].

Several interventions, including folic acid, betaine, osmotin, and adiponectin, demonstrated protective effects against ethanol-induced teratogenic effects and neuronal apoptosis [50,51,52]. Others, like DHA and papaverine, improved ethanol-induced deficits in behavior and neurodevelopmental markers [53,54]. Additionally, some interventions, such as HX106 and DAT inhibitor CE-123, improved enhanced cognitive abilities and regulated dopamine-related receptors in PAE animals [55,56].

These different interventions offer promising avenues for addressing the detrimental impacts of PAE on neurodevelopment and behavior, showing potential in mitigating or reversing ethanol-induced neurological deficits.

**Table 1 biomedicines-12-00496-t001:** Preclinical treatments in animal models of fetal alcohol spectrum disorders.

AuthorYearCountry	Pharmacological Intervention	Study Design and Objectives	Population	EtOH Exposure	Dose/Intervention Period	Variables Studied	Key Results
Patten et al. [43], 2013 Canada	Omega-3 fatty acids	RCT to determine whether feeding an omega-3 fatty acid-enriched diet from birth is able to overcome the deficits in synaptic plasticity that occur with PNE	Sprague-Dawley rats (N = 24)Offspring (N = 10 each group)	Prenatal exposure to EtOHOffspring exposure	Dams:‣on GD1: prenatal diets EtOH or Pair-fed or Ad libitum control‣on GD 15: BAC assessment. When pups were born, the dams were placed on either a regular chow diet or an omega-3-enriched dietPups:‣at P22: they were continued on The same postnatal diet as their mothers until they reached an experimental age (P55–70)‣at P70: animals were used for in vivo electrophysiological experiments‣Pre-stimulation: pulse 0.12 ms (duration) at 0.067 HzLTP induced by applying TBS‣Post-stimulation: 60 min at 0.067 Hz	EtOH, LTP, omega-3 fatty acids	omega-3 fatty acids can reduce oxidative stress and enhance antioxidant protection in PNE animals
Naseer et al. [52], 2014	Osmotin	In vitro and in vivo study to examine the neuroprotective effects of adiponectin-activated pathways against FAS	Rats Sprague-Dawley rats (P7)	exposed to EtOH for 12 and 24 hno prenatal exposure to EtOH	‣CT: no treatment‣TG1: EtOH (100 mM)‣TG2: Osm (0.16 mM) + ‣EtOH (100 mM) after 24 h staining with FJB and PI‣20% EtOH in normalsaline (4 mg/g body weight)	Osm, EtOH, caspase-3, AMPK activity	‣AMPK-activating drugs such as metformin, adiponectin, and osmotin prevent the effects of FAS‣Osmotin is an experimental alternative to adiponectin
Liang et al. [47], 2014 USA	Dihydromyricetin	RCT to study the effect of FAE on physiology, behavior, and GABAAR function of early adolescent rats to test the utility of DHM as a preventative treatment for FAE-induced disturbances	Sprague-Dawley young rats	Prenatal exposure to EtOHOffspring exposure	‣Dams: pregnant dams received the following treatments: (1) EtOH (1.5 g/kg, gavage); (2) EtOH (2.5 g/kg); (3) EtOH (5 g/kg); (4) EtOH (1.5 g/kg) + DHM (1 mg/kg); (5) EtOH (2.5 g/kg) + DHM (1 mg/kg); (6) EtOH (5.0 g/kg) + DHM (1 mg/kg); (7) DHM (1 mg/kg); (8) vehicle (water, 20 mL/kg)‣Offspring: Behavioral and electrophysiological experiments were performed on P25-35	EtOH; DHM; anxiety-related behavior; Plasma EtOH Concentration assessment; brain analysis	The absence of adverse side effects and the ability of DHM to completely prevent FAE consequences suggest that DHM is an attractive candidate for development as a treatment for the prevention of FASD
Bearer et al. [38], 2015 USA	Choline	RCT to determine if choline supplementation prior to an acute exposure to ethanol on P5 would reduce the effects of EtOH on the dowel crossing test	C57B16/J mice pups	No prenatal exposure to EtOH	Prenatal treatment from E 4.5 to P21: pups received choline-deficient pellet dietPostnatal treatment (from P2):Pre-exposure treatment: S group: 10 μLC group: 10 μL of 18/8 mg/mLExposure (on P5):EtOH group (6 g/kg) Intralipid^®^ group Post-exposure (from P6 to P20):S group: 10 μLC group: 10 μL of 18/8 mg/mL On P30: balance and coordination assessment (6 animals per sex per treatment group)	BAC; balance and coordination; Dowel Crossing Test	Choline fortification of common foodstuffs may reduce the effects of alcohol
Wellmann et al. [53], 2016USA	Docosahexaenoic acid (DHA)	RCT to evaluate if DHA reduction underlies (or contributes to) the neurobehavioral problems observed in FASD, to determine if postnatal DHA supplementation will ameliorate neurobehavioral problems in animal model of FASD	Long-Evans rats	Prenatal exposure to EtOHOffspring exposure	Dams: Timed pregnant rats were assigned to one of 3 groups: ‣ad libitum access to an ethanol-containing liquid diet;‣pair fed an isocaloric isonutritive non-alcohol liquid diet;‣ad libitum access to chow and water.Pups:From P11 to P20 pups were assigned to:‣DHA (10 g/kg in artificial rat milk);‣artificial rat milk 0.01 mL/g;‣CT.	P14: iUSVs assessment.P28 or P42: Social behavior and play-induced USVs assessment. P33 or P42: somatosensory performance assessment.P35: anxiety assessment.	DHA administration may have therapeutic value to reverse some of ethanol’s damaging effects
Balaraman et al. [39], 2017USA	Choline	RCT to examine whether the nutrient choline would modify ethanol’s effects on miRNAs	Sprague-Dawley rats (N = 48; 6 for each experimental group)	No prenatal alcohol exposure	From P4 to P21:‣EtOH (2.625 g/kg) + C (100 mg/kg/day) + 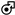 (or 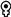 )‣EtOH (2.625 g/kg) + S + 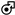 (or 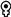 )‣Sham + C (100 mg/kg/day) + 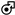 (or 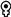 )‣Sham + S + 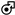 (or 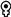 )	EtOH, choline, P22: brain examination, with dissection of the hippocampus and isolation of RNA	Choline modification of EtOH effects on miRNAs provide another avenue through which choline may protect against ethanol’s teratogenic effects
Waddell et al. [40], 2017USA	Choline	RCT to test if working memory training during adolescence combined with choline supplementation would improve cognitive flexibility in rat model of FASD more effectively than either intervention alone	Rats (GD3)	Prenatal exposure to EtOHOffspring exposure	Dams:‣CT group (N = 9) ‣ET group (N = 10): diet containing 3% ethanol for the duration of the exposureFrom P16 to P30:‣SC saline (0.9% Sal; pH 7.4) group‣SC choline (Cho; 100 mg/kg) groupFrom P30 to P38:‣Tr group ‣Utr group Offspring assigned to one of 4 postnatal groups: Sal-Utr; Sal-Tr; Cho-Utr; Cho-Tr	EtOH, Choline, working memory	Possible beneficial effect of choline in FASD; it may be more efficacious when coupled with cognitive training
Sogut et al. [50], 2017 Turkey	Folic acid and betaine	RCT to investigate prenatal EtOH exposure-related neuroapoptosis on the cerebral cortex tissues of newborn rats and possible neuroprotective effects of betaine, folic acid, and combined therapy	Sprague-Dawley ratsPregnant rats N = n.d.Offspring N = 65		Pregnant rats were divided into 5 experimental groups: control, EtOH, EtOH + betaine, EtOH + folic acid, and EtOH + betaine + folic acid combined therapy groups	Betaine, Calpain, cathepsin B and L, enzyme activities Caspase-3, Cytochrome c, Folic acid	Folic acid, betaine, and combined therapy of these supplements may reduce neuroapoptosis related to prenatal alcohol consumption and might be effective in preventing FAS
Wang et al. [57], 2018 USA	Minocycline	In vitro and in vivo study to demonstrate the neuroprotective role of minocycline	C57BL/6 Mice	No prenatal EtOH exposure	On PD5, pups were randomly assigned to: CG, minocycline group, EtOH group, and EtOH + minocycline	Minocycline, EtOH, microglial cells and neurons	minocycline may ameliorate EtOH neurotoxicity in the development by alleviating GSK3β-mediated neuroinflammation
Skorput et al. [58], 2019USA	NKCC1 inhibitor Bumetanide	In vivo and ex vivo study to assess whether the NKCC1 cotransporter is a tractable pharmacological target for normalizing in utero ethanol exposure-induced escalation of tangential migration to the prefrontal cortex	Nkx2.1Cre/Ai14 mice harboring Tomato-fluorescent (Nkx2.1+)	Dams EtOH exposureDams CT	‣5% *w*/*w* EtOH alone, from E13.5 to E16.5‣5% *w*/*w* EtOH + Bumetanide DMSO solution: 0.15 mg/kg/day, 3 days‣Isocaloric controldiet containing maltose	‣Density of Nkx2.1+profiles in the embryonic prefrontal cortex at E16.5‣PV+ interneurons in the PFC of young adult mice exposed to EtOH and EtOH + Bumetanide	‣EtOH alone ↑ Density of Nkx2.1+ neurons‣Bumetanide ↓ ethanol-induced enhancement of tangential migration of GABAergic cortical interneurons vs. CT
Ren et al. [59],2019USA	Minocycline	RCT to determine whether minocycline’s neuroprotective and anti-inflammatory properties can reduce EtOH-induced damage to spinal cord neuron development	C57BL/6 mice	No prenatal EtOH exposure	On PD5:‣CT (SC injections of sterile saline)‣Min group (2 SC injections/30 mg/kg)‣EtOH group (5 g/kg)‣EtOH (5 g/kg) + Min (2 SC injections/30 mg/kg)BAC = 338 mg/dL 8 h after the first injection	EtOH, minocycline, spinal cord	Minocycline may have neuroprotective property against ethanol-induced spinal cord damage. More investigation of the dosage and timeline of minocycline’s action is necessary.
Montagud-Romero et al. [49], 2019	Trichostatin A (TSA)	RCT to assess the effects of TSA on emotional and cognitive impairments caused by PLAE	C57BL/6 mice	Prenatal exposure to EtOH (N = 30)Offspring (N = 80 males)	Dams: binge alcohol drinking model during gestation and lactation (20% *v*/*v* alcohol solution)Pups:From P28 to P35: treatment with TSA from P36 to P55 behavior assessment	EtOH, TSA, anxiogenic-like responses, and memory deterioration	potential benefits of HDAC inhibitors for some aspects FASD
Cantacorps et al. [46], 2020Spain	Curcumin	RCT to assess the effects of curcumin in counteracting behavioral and molecular alterations induced by EtOH exposure during development	C57BL/6 mice (N = 24 for PAE)	Prenatal exposure to EtOHOffspring	Dams:‣DID test‣CG: waterEG: 20% EtOH (oral)Pups:‣P28: daily treatment with curcumin and vehicle solution‣P60: EPM‣P61: Y-maze‣P62: NOR‣P66: T-maze‣P70: brain extraction	Curcumin; anxiety-like behavior; working memory; recognition memory; spatial ability	Curcumin could be a useful treatment for attenuating the alcohol-induced deficits observed in FASD
Cadena et al. [51], 2020USA	Folic Acid	RCT to test FA ability to protect against behavioral defects induced by EtOH exposure	zebrafish	No prenatal exposure to EtOH	Healthy embryos were maintained in embryo medium until 2 hpf ‣CT group: animals transferred to embryo medium, no treatment ‣EtOH group: embryo medium containing 100 mM EtOH‣EtOH + FA group: embryo medium containing 100 mM EtOH and 75 μM FA‣FA group: embryo medium containing 75 μM FA	FA, EtOH, Teratogenic effects on morphological development (COA, ED, EY, PE, SB, SD, TD, YSE, YSR, HR, LP)Behavior (locomotor activity, sleep, anxiety like-behavior, visual ability)	FA alone produced different behavioral responses, indicating that FA may independently modify fish development. This zebrafish behavior on FASD model also provides an opportunity to test whether other compounds reduce the EtOH toxic effects.
Bottom et al. [41], 2020USA	Choline	RCT to assess the ability of concurrent choline supplementation to ameliorate atypical neocortical and behavioral development following fully or partially PAE	CD-1 Mice	Prenatal exposure to EtOHOffspring	Dams (N = 10 for each group):‣group 1: Water (Control)‣group 2: 25% EtOH in water ‣group 3: 25% EtOH in water with 642 mg/L choline chloride‣group 4: 642 mg/L choline chloride in water (CW).Separate subsets of dams were sacrificed at either GD9 and 19 for BEC and POSM measurements during gestation (n = 10, all groups).Pups:‣Experimental analyses in brain: 1 ± 1 pups (P0) were selected pseudo-randomly from each litter.‣Behavioral analysis: subsets of each litter in all groups were cross-fostered to alcohol-naïve mothers until P20 when behavioral testing of 3 ± 1 pups per litter took place.	EtOH, Choline, BEC, POSM, anxiety-like behaviors, motor function	Choline supplementation may represent a potent preventative measure for the adverse outcomes associated with PAE
Mohammad et al. [60], 2020 USA	Kcnn2 blockers	In vivo study to assess if PAE leads to deficits in gross and fine motor skill learning	Mice	Prenatal exposure to EtOH	Dams exposed to EtOH (1.0 g/kg weight), at E 16 and 17, during which upper cortical layer neurons are predominantly generated	Kcnn2, motor cortex, motor learning deficits	Kcnn2 blockers may be a novel intervention for learning disabilities in FASD
Ju et al. [1], 2020Republic of Korea	HX106	RCT to assess the effects of HX106 on PAE-induced ADHD-like phenotype and metabolic changes in the offspring mice	ICR mice (28–30 g weight)	Prenatal exposure to EtOHOffspring	Female at GD 3.Stabilized animals divided into 3 groups (from GD 6 to GD 15):‣CT: saline (0.9% NaCl)‣EtOH 2 groups: 6 g/kg/day; 50 *v*/*v*%)Pups (from P21; n = 10 per each group)‣CT group: saline‣OPAE groups: saline + HX106 at 200 mg/kg/day	ADHD-Like Symptoms, HX106, EtOH	ADHD pathology may involve metabolic disruptions and HX106 can be a promising supplement to attenuate the hyperactivity appearance in PAE-induced ADHD-like neuropathology
Almeida-Toledano et al. [61], 2021Spain	Epigallocatechin Gallate	RCT to describe the effect of EGCG administration on oxidativestress, fetal growth, placental development, and neurogenesis processes in 2 human-like patterns of alcohol use FAS mouse models	283 mouse fetuses		‣Group 1: 42 Med control‣group 2: 47 Med EtOH‣group 3: 45 EtOH Med + EGCG‣group 4: 54 Bin control ‣group 5: 44 Bin EtOH ‣group 6: 47 Bin + EGCG EtOH	Blood EtOH levels; EGCG; fetal growth; EtOH effect on oxidative stress; neuronal maturation and plasticity; astrocyte differentiation	EGCG is a promising antioxidant therapy to attenuate the consequences of PAE
Gibula-Tarlowska et al. [56], 2021 Poland	CE-123 DAT inhibitor	RCT to demonstrate that dopamine signaling is one of the main factors underlying hyperactive, inattentive, and impulsive behaviors	Male rat (N = 163)	No prenatal exposure to EtOH	EtOH on P4-9.‣Weaning: P20 group 1 was tested in early adolescence (P21)‣group 2 was tested as adults (P45-50)group 3 was tested at P50	BEC; Locomotor activity test; EPM test; Barnes maze task	CE-123 may be helpful in overcoming behavioral disorders in subjects perinatally exposed to ethanol
Lopatynska-Mazurek et al. [45], 2021Poland	Rapamycin	Prospective cohort study to evaluate the hypothesis that development of emotional learning deficits and depressive-like behaviors in adult rats exposed to EtOH during the neonatal period are a function of oxidative stress that, in turn, depends on the mTOR signaling pathway	Male rats (N = 64)	No prenatal exposure to EtOH	from P4 to P9:‣S (0.9%) + SI‣Rapamycin + SI‣Torin 2 + SI‣FK-506 + SI‣S (0.9%) + EtOH (5 g/Kg/day)‣Rapamycin + EtOH (5 g/Kg/day)‣Torin 2 + EtOH (5 g/Kg/day)‣FK-506 + EtOH (5 g/Kg/day)‣P60 (adults): aversive learning and memory processes assessment; depressive-like behavior assessment	EtOH, Rapamycin, Torin, FK-506	Rapamycin but not Torin-2 or FK-506 could have protective effects against the ethanol-induced LPO and AP site levels in the hippocampus and prefrontal cortex.Rapamycin could be useful as a preventive therapy in disorders related to PAE.
García-Baos et al. (2021) [62]Spain	Cannabidiol	RDBPC to explore CBD therapeutic effects demonstrating that it might reduce through an anti-inflammatory mechanism. Cognitive deficits induced by EtOH exposure.	C57BL/6 mice	Prenatal exposure to EtOHOffspring	Pregnant females: DID test during PLAE. Day 4: six binge-like drinking sessions. Dose ethyl alcohol n.d.Offspring: CBD 20 mg/kg during 10 consecutive days from P25 until P34Behavioral assessment: P60–P66P70: PFC and HPC extraction	CBD, EtOH, spatial, working and recognition memory	CBD appears to be a promising therapeutic drug since it could hamper cognitive impairments caused by several pathological conditions, potentially through neuroimmune modulation
Grafe et al. [42], 2021Canada	Choline	RCT to examine the impact of acute choline administration on synaptic transmission in the in vitro DG and how these mechanisms may be altered by EtOH exposure of male and female offspring	Sprague-Dawley rats		GD1 2 experimental groups: ‣PNE diet was a protein and liquid diet containing 35.5% ethanol-derived calories. Dams were gradually introduced to the ethanol diet and subsequently consumed only the ethanol diet from GD 22 until the day before birth. ‣On GD 21: dams were returned to a solid control chow diet and remained on this diet until parturition. CT diet.	EtOH, choline, NMDA, M1 receptors	choline can modulate hippocampal transmission at the level of the synapse and it can have unique effects following EtOH exposure
Sharma et al. [2], 2021India	Papaverine	RCT to assess the outcome of papaverine administration on PAE affected behavioral (hyperactivity, repetitive behavior, and anxiety) and biochemical markers in several brain areas associated with ADHD	N = 40 Adults albino wistar rats	Prenatal exposure to EtOHOffspring	Dams (from GD8 to GD20): ‣EtOH (20% *w*/*v*; 6 g kg^−1^ day^−1^-weekdays; 4 g kg^−1^ day^−1^-weekends; N = 8);CT group: received calorifically equivalent sucrose solution (30% *w*/*v*; N = 5)Pups (from P21 to P48), 4 Exp groups (each with 8 males):‣Group I and II (VEH/Drug per-se); sucrose solution-treated females were administered with normal saline—2 mL kg^−1^ i.p., or Papaverine—30 mg kg^−1^ day^−1^, i.p., as per the treatment assigned.‣Group III (PAE); females treated with EtOH during gestation were treated with NS—2 mL kg^−1^, i.p.‣Groups IV and V (Papaverine treatment); PAE pups were divided further into PAE + P (15/30) groups and received papaverine (15/30 mg/kg day^−1^ i.p.) in a volume of 2 mL kg^−1^.‣Behavioral assessment: P44 to P48.	ADHD-Like Symptoms; papaverine;EtOH	Papaverine, a selective PDE10A inhibitor rectified behavioral phenotypes associated with ADHD, possibly by altering the protein markers associated with neuronal survival, neuronal transcription factor, brain inflammation, and brain oxidative stress. Implicating PDE10A as a possible target for furthering our understanding of ADHD phenotypes.
Chen et al. [63], 2022Taiwan	AST	RCT to explore how AST treatment can ameliorate morphological changes in the hippocampus and cognitive impairment in FASD rats by reducing oxidative stress and neuroinflammation in the brain	40 male rat pups	No prenatal exposure to EtOH	4 groups: CT; normal with AST; FASD group; ethanol-inhaled with AST treatment. FASD induction started on P2 and continued until P10. To investigate AST effects in a FASD rat model, it was administered on P53 and kept until sacrifice (P60).	EtOH, AST, MWM task, BWT	AST could be considered to treat FASD
Sabzali et al. [44], 2022Iran	Metformin	In vivo and ex vivo study to evaluate the protective effects of metformin on EtOH-related neuroinflammation, as well as neuron apoptosis in the hippocampus of adult male rat model of FASD	60 Wistar male rat pups (8–10 g)	No prenatal exposure to EtOH	EtOH in milk solution (5.25 and 27.8 g/kg, respectively) by intragastric intubation at 2–10 days after birth	tumor necrosis factor-α (TNF-α) and antioxidant enzyme concentrations	Metformin reducescell apoptosis in the hippocampus. It can be considered a promising therapeutic option for FASD; however, more research is required.
Burton et al. [64], 2022 USA	Smoothened Agonist (SAG); purmorphamine (PUR)	In vivo study to demonstrate that a Shh pathway agonist given at an appropriate dose and timing relative to ETOH exposure during embryogenesis may alleviate the effects of PAE	Zebrafish embryos (AB strain, ZFIN ID: ZDB-GENO-960809-7)	Zebrafish embryos exposed to EtOH from 8–10 hpf	Embryos exposed to varying concentrations of SAG or PUR at 6–8 hpf or 10–12 hpf by diluting with egg water	EtOH, Eye size data, gene expression, novel tank diving data, midbrain hindbrain data, and pax6a in situ expression data	Pharmacological activation of the Shh pathway at specific developmental timing markedly diminishes the severity of alcohol-induced birth defects as altered eye size and midline brain development
Farhadi et al. [65], 2022Iran	Crocin	RCT to evaluate the protective impact of crocinon ethanol-related neuroinflammation	72 wistar rat pups (8 to 10 g)	No prenatal exposure to EoTH	Pups treated from P2 to P10.Group 1: CTGroup 2: milk + saline solution (dose n.a.)Group 3: milk solution + ethanol (total daily dose of ethanol (5.25 g/kg)Groups 4, 5, 6: milk solution + ethanol + crocin (15, 30, and 45 mg/kg)	EtOH; Crocin; astrogliosis evaluation; inflammation measurement; enzyme evaluation; spatial memory	Crocin is applicable for the treatment ofFASD
Gasparyan et al. [66] 2023Spain	Cannabidiol	RCT to evaluate the effects of early and chronic CBD administration on offspring exposed to an animal model of FASD	N = 190 c57bl/6j mice50 female and 20 male 5-week-old mice Offspring:120 (60 males and 60 females)	Prenatal exposure to EtOHOffspring	PAE: N = 15 females exposed to 2 bottles with tap water.N = 35 females exposed to 2 bottles, one always with tap water and the other increasing EtOH concentrations.PAE: at GD7, EtOH gavage was started until the pup’s weaning at P21.Pups: treated with CBD 30 mg/kg/day or VEH from the day of weaning for 10 weeks.	CBD, ethanol, anxiety-like behaviors, depressive-like behaviors; ASR; recognition memory; long-term aversive memory; gene expression analyses; brain analysis	Potential suitability of CBD in children and young adults with FASD

**Abbreviations:** 
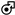
, male gender; 
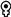
, female gender; ADHD, attention deficit hyperactivity disorder; AP, apurinic/apyrimidinic; AST, astaxanthin; EC or BAC, blood ethanol concentration; Bin, binge; BWT, beam walking test; C or Cho, choline; CBD, cannabidiol; Cho-Tr, choline-trained; Cho-Utr, choline-untrained; COA, coagulation; CT, control group; DAT, dopamine transporter; DHA, docosahexaenoic acid; DHM, dihydromyricetin; DG, dentate gyrus; DID, drinking in the dark; E, embryonic day; ED, large inter-eye distance; EGCG, epigallocatechin-3-gallate; EPM, elevated pus maze test; ET, ethanol-exposed group; EtOH, ethanol or alcohol; EY, small eyes; FAE, Fetal alcohol exposure; FAS, fetal alcohol syndrome; FJB, fluoro-jade B; FST, forced swimming test; GABAAR, γ-aminobutyric acid type A receptor; GD, gestational day; GFAP, glial fibrillary acidic protein; GSH-Px, glutathione peroxidase; GSK3β, glycogen synthase kinase 3 beta; HAT, histone acetyltransferase; HDAC, histone deacetylases enzymes; HPC, hippocampus; hpf, hour post-fertilization; HR, hatching rate; larvae paralysis; iUSVs, isolation-induced ultrasonic vocalizations; LPO, lipid peroxidation; LTD, long-term depression; LTP, long-term potentiation; Med, mediterranean; Min, minocycline; mIPSCs, inhibitory postsynaptic currents; mTOR, mammalian target of the rapamycin; MWM, morris water maze; n.d., not declared; NMDA, N-methyl-d-aspartate; NOR, novel object recognition test; OPAE, Offspring prenatal alcohol exposure; Osm, osmotin; P or PD, postnatal day; PAE, prenatal alcohol exposure; PE, pericardial edema; PFC, prefrontal cortex; PI, propidium iodide; PLAE, prenatal and lactational alcohol exposure; PNTG, postnatal treatment group; PNUn, postnatal untreated group; POSM, plasma osmolality; PUFAs, polyunsaturated fatty acids; RCT, randomized controlled trial; RDBPC, randomized double blind placebo control study; S, saline; Sal-Tr, saline-trained; Sal-Utr, saline-untrained; SB, delayed swim bladder inflation; SC, subcutaneous; SD, spine deformation; Shh, sonic hedgehog; SI, sham intubation; SOD, superoxide dismutase; TBS, theta burst stimulation; TD, tail deformation; TG, treatment group; THIP, gaboxadol hydrochloride; TNF-α, tumor necrosis factor-α; Tr, working memory training; TSA, trichostatin A; Utr, untrained; VEH, vehicle; YSE, yolk sac edema/unusual large yolk sac; YSR, delayed yolk sac resorption.

### 3.3. Results from Human Studies

As reported in Table 2, most of the clinical studies carried out in the last ten years concerned both perinatally [67,68,69,70] and postnatally [71,72,73,74,75,76] the use of choline and/or vitamins as nutritional interventions for ND due to alcohol exposure. After the promising results obtained in animal studies, tolerability, adherence and side effects of choline were evaluated in humans, when perinatally and postnatally administered for approximately 6 and 9 months to pregnant alcohol users and 2–5 year-old children, respectively. These studies [68,71] demonstrated a good feasibility and acceptability of choline both in heavy drinking pregnant women and children with minimal or no side effects (details in Table 2).

When exploring choline efficacy on cognitive development of PAE children, two studies from the same research group [72,75] observed potentially positive effects on memory functions in children aged 2–4 years old, while a little decrease in behavioral problems was assessed at 8 years old. These results highlighted the need to perform long-term follow-up for the monitoring of choline effects on neurodevelopmental functions in PAE children. Another study with choline [73] did not obtain significant cognitive improvements in school-aged children. Indeed, according to preclinical studies, choline supplementation has greater efficacy in early ages and a choline therapeutic window for FASDs children has to be considered in clinical studies. Nevertheless, even when prenatally administrated, choline did not minimize PAE effects, when measured at 6 months of age and only the use of multivitamins had small positive effects [67].

Warton et al. reported that high doses of choline supplementation (2 g/day) can mitigate PAE-related brain structural deficits in humans. Gimbel et al. [76] obtained similar results in a follow-up study, but with a lower dose. The authors showed that early choline supplementation may affect white matter development in terms of the structure and organization of axons in the corpus callosum.

**Table 2 biomedicines-12-00496-t002:** Pharmacological interventions for fetal alcohol spectrum disorders.

Author Year Country	Study Design and Objectives	Pharmacological Intervention	Dose Intervention Period	Population (N)	Patient Age, Mean (SD)	Variables Studied	Key Results/Assessment
Wozniak J.R. et al. [71], 2013 USA	RDBPC to evaluate tolerability and bioavailability of choline as supplement for FASD children	Postnatal choline bitartrate (1.25 g)	514 mg/day, 9 m	Children: ‣10 active ‣9 placebo	2–5 y (range)	‣Feasibility ‣Tolerability ‣Adverse effects ‣Serum choline levels	‣Minimal and equivalent adverse effects were observed in choline and placebo groups. Fishy body odor was reported for choline group ‣Acceptable feasibility and tolerability
Wozniak J.R. et al. [72] 2015 USA	RDBPC to assess hippocampal-dependent memory improvement by postnatal choline supplementation in FASD children	Postnatal choline bitartrate (1.25 g)	514 mg/day, 9 m	Children: ‣31 active ‣29 placebo	3.8 y (0.80)	‣Mullen Scales of Early Learning (global cognitive functioning) ‣Elicited imitation memory tasks	‣No choline effects were observed in global cognitive functioning ‣Choline ↑ hippocampal-dependent memory tasks especially in younger children (≤4 y) and best score were obtained in long-delayed memory
Coles C.D. et al. [67], 2015 USA Ukraine	PCS to evaluate mother nutritional supplements use impact on children prenatally exposed to alcohol	‣prenatal MVM ‣prenatal MVM + Choline	‣nd ‣nd + 750 mg Choline from first prenatal clinic visit until delivery	367 Children		‣Children development assessment through BSID-II at 6 m of age	‣MVM ↑ cognitive development ‣Choline has no effect on cognitive development and negative effects on motor outcomes ‣No differences between prenatally alcohol-exposed and non-exposed children
Nguyen T.T. et al. [73], 2016 USA	RDBPC to investigate if choline supplements have positive effects on the memory impairments, executive function, and attention deficits in school-aged (5–10 y) FASDs children	Postnatal Glicerophospho-choline	625 mg/day, 6 wk	55 Children: ‣29 active ‣26 placebo	8.3 y (1.75)	‣Cognitive abilities in the domains of learning and memory, executive function, (attention, fine motor functioning)	‣Choline supplements did not improve memory, executive and attention functioning in school-aged (5–10 y) children with FASDs
Jacobson S.W. et al. [68], 2018 South Africa	RDBPC to assess adherence and side effects to choline supplementation in heavy drinking pregnant women	Prenatal choline bitartrate	2 g/day, from mid-pregnancy until delivery	Mothers: ‣34 active ‣35 placebo	26.4 y (5.7)	‣Adherence ‣Side effects ‣Choline plasma concentration	‣Choline adherence was excellent (≥84%) and good to excellent (≥68%) for 42% and 58% of participants, respectively. It was not related to maternal education, depression, intellectual function, stressful life events or socioeconomic status. ‣Increase in nausea/dyspepsia symptoms, especially when choline is consumed on an empty stomach. ‣Choline levels in plasma were significantly higher than those of the placebo group.
Jacobson S.W. et al. [69], 2018 South Africa	RDBPC to assess prenatal choline supplementation effects on EBC, on growth, recognition memory and information processing speed deficit	Prenatal choline bitartrate	2 g/day, 9–27 wk (range)	62 Infants: ‣31 active ‣31 placebo	‣6.5 m ‣1 y	‣EBC assessed at 6.5 m ‣Infant growth assessed at 6.5 m and 1 y ‣Visual recognition memory assessed at 6.5 m and 1 y through FTII ‣FASD diagnosis	‣Prenatal choline supplementation ↑ conditioned responses more than placebo, especially in mother with high adherence to the treatment. ‣Newborns in the placebo and choline group were both small at birth. Infants in choline group; nevertheless, a weight and head circumference growth increase between 6.5 m and 1 y ‣FAS and pFAS diagnosis incidence for the choline and placebo group was almost the same
Sarkar et al. [74], 2019 USA	RDBPC to assess if choline can reduce DNA methylation and improve POMC and PER2 gene expression by analyzing blood sample of FASD children treated with the supplement	Postnatal choline bitartrate	514 mg/day, 9 m	32 Children: ‣16 active ‣16 placebo	2.5–5 y (range)	‣POMC and PER2 mRNA levels measured by quantitative real-time PCR ‣DNA methylation assessed by methylation-specific real time PCR assay	‣Choline increased POMC and PER2 gene expression following 9 m of treatment ‣Choline reduced PER2 and POMC DNA methylation following 9 m of treatment
Wozniak J.R. et al. [75], 2020 USA	RDBPC follow-up study evaluating postnatal choline supplementation potential on long-term cognitive and behavioral functions of PAE children	Postnatal choline	514 mg/day, 9 m	31 Children: ‣15 active ‣16 placebo	8.6 y (1.0)	‣Cognitive, memory, executive, behavioral and emotional functioning	‣Significant non-verbal Visual-Spatial Reasoning and non-verbal Working Memory components in the choline group vs. placebo ‣choline had non-significant effects on quantitative reasoning, fluid reasoning, and a range of verbally-mediated skills ‣Choline group had fewer ADHD-related behavioral problems than the placebo group
Warton F.L. et al. [70], 2021 South Africa	Follow-up study observing if prenatal choline supplementation protected PAE newborn brain from volume reduction and improved infant recognition memory	Prenatal choline	2 g/day	50 Children: ‣27 active ‣23 placebo	2.8 wk median	‣MRI data of newborn brains ‣Recognition memory through FTII at 12 months	‣Larger total intracranial volume in choline group vs. placebo ‣Larger right putamen and corpus callosum were associated with improved recognition memory
Gimbel B.A. et al. [76], 2022 USA	Follow-up study evaluating postnatal choline effects on executive functioning and white matter microstructure	Postnatal choline bitartrate	514 mg/day, 9 m	18 Children: ‣9 active ‣9 placebo	11.0 y	‣Executive functioning ‣Differences in corpus callosum and white matter microstructure through diffusion-weighted MRI and the NODDI biophysical model	‣Improved executive functioning in choline group vs. placebo ‣Lower corpus callosum orientation dispersion index (more coherent fibers) in choline group vs. placebo

**Abbreviations: ↑**, improve; **ADHD**, attention deficit/hyperactivity disorder; **BSID-II**, Bayley Scales of Infant Development 2nd Ed; **EBC**, eyeblink conditioning; **FASDs**, Fetal Alcohol Spectrum Disorders; **FAS**, Fetal Alcohol Spectrum; **FTII**, Fagan Test of Infant Intelligence; **m**, months; **MRI**, magnetic resonance imaging; **MVM**, multivitamin/mineral supplementation; **nd**, not declared; **NODDI**, neurite orientation dispersion and density imaging; **PAE**, prenatal alcohol exposure; **PCS**, prospective cohort study; **pFAS**, partial fetal alcohol spectrum; **RDBPC**, randomized double blind placebo control study; **wk**, weeks; **y**, years.

## 4. Discussion

PAE has long been recognized as a significant risk factor contributing to FASD. Early diagnosis plays a crucial role in addressing FASD symptoms and minimizing subsequent SD that may arise later in life.

FASD is associated with various comorbidities [24], significantly impacting the individual’s lifelong health and functionality. ND, such as Cognitive Delay, ADHD, SLD, and Externalizing Disorders, are commonly observed in FASD subjects.

The therapeutic approach for FASD individuals requires a multidisciplinary strategy, including medical, psychosocial, and educational interventions [19].

Several drugs have shown potential in managing FASD-related symptoms. Mood stabilizers, CNS stimulants, atypical antipsychotics, SSRIs, benzodiazepines, and other medications have been used to address aggression, ADHD, intellectual disabilities, and depressive or anxiety disorders. However, it is crucial to consider potential side effects and long-term impact, especially given the teratogenic effects of certain drugs.

Stimulants are primarily used to counteract ADHD symptoms including hyperactivity, inattention, and impulsivity. For example, the use of methylphenidate in a pilot study showed improvements in the symptoms of ADHD although there are opinions against its use due to side effects (decreased appetite, stomach aches and headaches, growth retarding effects, diversion) [77]. Doing et al. [78] reported contradictory findings about the use of methylphenidate in cases of comorbidities associated with FASD (symptoms of hyperactivity and inattention). Even considering study limitations, inattention was found to respond better to dextroamphetamine.

If stimulants are not effective, selective norepinephrine reuptake inhibitors (atomoxetine) may be used to treat ADHD symptoms. Anxiolytics can treat anxiety symptoms commonly seen in children with FASD. Alpha-2-adrenergic agonists are used to counteract the symptoms of anxiety and have also been shown to address problems with attention, hyperactivity, and impulsivity, as well as helping to reduce tics. Because alpha-2-adrenergic agonists can cause sedation, their daytime use may be limited, but they may be helpful for children who have difficulty sleeping. Antidepressant medications can treat depressive symptoms such as sad mood, loss of interest, sleep problems, and anxiety. Neuroleptics (for example, risperidone or aripiprazole) may be used to treat serious symptoms such as aggression, anxiety, disruptive behavior, tics, and impulsivity. The most common side effect of neuroleptic use is significant weight gain, and rarely the drug can cause extrapyramidal seizures. Melatonin supplementation can help with sleep disorders. Some subjects with FASD who exhibit sleep difficulties may respond positively to melatonin supplementation to aid sleep onset [79].

Moreover, when evaluating individuals by a Social Skills Rating System, in a Children’s Friendship Training treated with psychotropic drugs, different outcomes have been recorded. For example, neuroleptics improved Social Skills Rating System scores (parent- and teacher reported self-control, parent assertation and problem behaviors) more than stimulants, while antidepressants and nonstimulants did not produce statistically significant main outcomes. In some cases, no outcome differences between children prescribed stimulants or no medication were described [80].

The choice of a pharmacological intervention for treating FASD depends on many factors such as the severity of the disease spectrum and on gender and age. Although some clinical evidence has confirmed the validity of different therapy approaches, there are still many studies needed to determine the effectiveness of appropriate treatment [81].

In the past 10 years, new experiments have been conducted with new substances, especially non-psychotropic molecules.

This integrative review shows that in the last 10 years, the focus has changed from treating symptoms to prevention. The new animal and human studies have emphasized the importance of testing substances that can affect the embryogenesis of the various brain regions altered by alcohol exposure at the subcellular level. Acting during embryogenesis on the negative effects of alcohol like neuronal apoptosis, synaptic connections, cascade of protein signaling, and morphologic modifications, could reduce PD and ameliorate SD. Alternative interventions in animal studies such as natural antioxidants like EGCG and cannabidiol have shown promise in ameliorating FASD-related cognitive impairments or addressing associated symptoms.

For example, Epigallocatechin Gallate (EGCG) has been suggested to be a protective agent against FASD, ameliorating fetal growth restriction and preventing FASD-related cognitive impairment [48].

The majority of the trials are still in preclinical stages, and choline has also been studied in clinical trials. Despite the results of the studies agreeing on choline’s beneficial effect such as nonverbal intelligence, visuospatial abilities, verbal memory, and working memory, there are disparate conclusions regarding its use. There are still some issues to be addressed like establishing the right timing and doses of administration (both prenatally and postnatally). Moreover, further follow-up studies need to be performed to evaluate its impact on older age. When thinking about this treatment approach (both during the prenatal and postnatal period), the early diagnosis is of fundamental importance not only in managing SD, but also in PD, even though the biggest limit in perinatal cases is the detection of pregnant women who declare that they drink.

The selected studies encompassed human and animal models, evaluating pharmacological treatments’ efficacy on neurocognitive and behavioral responses in FASD patients.

However, limitations include the varied study designs and small population sample size with limited randomized trials using novel molecules, hindering the generalization of results.

The review strengths lie in its comprehensive assessment of the new insights in pharmacological therapies tested for children with FASD with diverse studies, including the treatment with promising nutritional supplements as choline and EGCG. Moreover, another strength of this study is that it highlights key results from previous papers using a snowball strategy to cover all the known treatments for FASD comorbidities.

As a limitation, this review did not evaluate the quality of evidence following the GRADE statement, which may lead to bias in relevance or specific value among the different studies. Furthermore, FASD is associated with several comorbidities, which complicates the search strategy. Hence, we decided to focus on the most representative comorbidities associated with FASD.

## 5. Conclusions

In conclusion, up to date, there are specific treatments for FASD-associated symptoms caused by the cellular and tissue injury that originated from prenatal exposure to alcohol, but there are no medicaments addressed to an irreversible damage. However, recent studies included in this review have performed multidisciplinary approaches involving psycho-social interventions and pharmacological treatments that aim to improve FASD individuals’ quality of life and manage its associated symptoms. Moreover, natural molecules such as choline or EGCG have shown promising results in recent clinical studies; therefore, they could be used as a complementary treatment for some of the clinical manifestations in subjects with FASD. However, further research and longer-term follow up need to be performed to better understand the efficacy, safety, and long-lasting impact of these interventions in FASD individuals. Furthermore, the restorative therapies that have been used in both animal and human models have to be aimed to improve damaged neural functioning. An example is antioxidants that can stabilize neural membranes, in a similar way as the treatment of fever associated with meningitis or pneumonia. The authors believe that this should be the main investigation line for studies in the next future.

This integrative review serves as a crucial reference for clinicians and researchers seeking insights into the novel approach of pharmacological interventions for FASD and associated comorbidities, driving the need to investigate in more targeted and effective treatments for affected individuals.

## Figures and Tables

**Figure 1 biomedicines-12-00496-f001:**
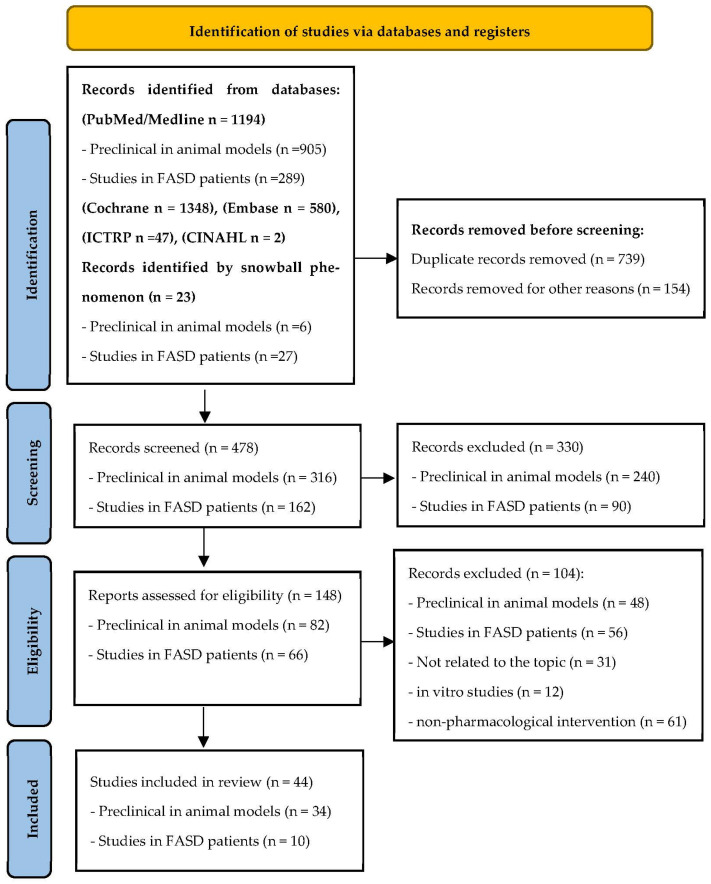
Methodological flowchart based on PRISMA 2020 update following the preferred reporting items for systematic review [36].

## Data Availability

Data are contained within the article.

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
