# Peer review of "Novel Interventions on Comorbidities in Patients with Fetal Alcohol Spectrum Disorder (FASD): An Integrative Review"

_biomedicines, 2024, doi:10.3390/biomedicines12030496_

Round 1
Reviewer 1 Report
Comments and Suggestions for Authors
The quality of the English needs to be improved in both the introduction and the discussion. Please have a native English speaker edit and assist in re-writing the introduction and discussion. Otherwise this paper is a very solid contribution to the literature: the research methods are good, the tables are well done and informative, and I agree with the conclusion that this review will serve as a "crucial reference for clinicians and researchers..."
Comments on the Quality of English LanguageThe quality of the English needs to be improved in both the introduction and the discussion. Please have a native English speaker edit and assist in re-writing the introduction and discussion.
Author Response
Reviewer 1: The quality of the English needs to be improved in both the introduction and the discussion. Please have a native English speaker edit and assist in re-writing the introduction and discussion. Otherwise this paper is a very solid contribution to the literature: the research methods are good, the tables are well done and informative, and I agree with the conclusion that this review will serve as a "crucial reference for clinicians and researchers..."
The quality of the English needs to be improved in both the introduction and the discussion. Please have a native English speaker edit and assist in re-writing the introduction and discussion.
As suggested by the reviewer, we improved the English in the introduction and the discussion sections as highlighted in red.
Reviewer 2 Report
Comments and Suggestions for Authors
The topic of paper is of great interest. The paper is well structured and facilitates the reading and the follow-up of the study. However, some clarifications must be included to accept their publication.
Keywords must comply with the Medical Subject Headings (MeSH).
A bibliographic review is a synopsis that summarizes different research and articles that gives us an idea of the current state of the issue to be investigated. In the review, a critical assessment of other research on a given topic is carried out, a process that helps us put the topic in its context.
In the review, the different methodologies and designs are evaluated, identifying advantages, disadvantages and difficulties that each methodological orientation presents. The purpose of the literature review is to make use of criticism and previous studies in an orderly, precise and analytical manner. In short, the literature review is presented as a critical analysis of the topic of interest while pointing out similarities and inconsistencies in the literature analyzed.
The review is a retrospective activity that provides us with information limited to a specific period of time. The authors must explain the reasons why the review starts from 2013.
One of the aspects that characterizes the integrative review is that a series of criteria are established in a transparent manner that ensure the quality of the results of the review.
Was any quality evaluation instrument applied to the selected studies?
The first three paragraphs of the discussion section are reiterations of comments already included in the introduction section. This information can be synthesized and accompanied by references.
The authors must include the limitations of the review carried out, the methodological gaps and the strong points of the review carried out.
The conclusions section is poor. This section should be expanded to include a more robust explanation of the results obtained.
Author Response
The topic of paper is of great interest. The paper is well structured and facilitates the reading and the follow-up of the study. However, some clarifications must be included to accept their publication.
- Keywords must comply with the Medical Subject Headings (MeSH)
We thank the reviewer for his valuable suggestion. Keywords were indexed using the Medical Subject Headings for greater compatibility with other information sources.
- A bibliographic review is a synopsis that summarizes different research and articles that gives us an idea of the current state of the issue to be investigated. In the review, a critical assessment of other research on a given topic is carried out, a process that helps us put the topic in its context. In the review, the different methodologies and designs are evaluated, identifying advantages, disadvantages and difficulties that each methodological orientation presents. The purpose of the literature review is to make use of criticism and previous studies in an orderly, precise and analytical manner. In short, the literature review is presented as a critical analysis of the topic of interest while pointing out similarities and inconsistencies in the literature analyzed.
We would like to thank the reviewer for appreciating the literature critical analysis performed on the topic by the authors.
- The review is a retrospective activity that provides us with information limited to a specific period of time. The authors must explain the reasons why the review starts from 2013.
The authors thank the reviewer for the valuable comment. The main aim of this review was to update the novel treatments applied to FASD comorbidities. For this reason, we decided to investigate these treatments in the last decade. Papers before 2013 were not included because mainly focused on the PAE effect investigation rather than research based on novel treatments for comorbidities associated to FASD. as we reported at lines 163-165
- One of the aspects that characterizes the integrative review is that a series of criteria are established in a transparent manner that ensure the quality of the results of the review. Was any quality evaluation instrument applied to the selected studies?
The application of quality evidence methodology such as GRADE, STROBE, CASPE was not possible considering our inclusion and exclusion criteria (human vs animal and in each model different treatments with different outputs variables). This is the reason why we called our review “an integrative review” as we did not perform a systematic review, which would have required evaluation instruments.
- The first three paragraphs of the discussion section are reiterations of comments already included in the introduction section. This information can be synthesized and accompanied by references.
The discussion section was modified according to the reviewer suggestion.
- The authors must include the limitations of the review carried out, the methodological gaps and the strong points of the review carried out.
As suggested, the authors highlighted the strengths, the limitations and methodological gaps at lines 383-397
- The conclusions section is poor. This section should be expanded to include a more robust explanation of the results obtained.
Following reviewer comments, conclusion section was modified mentioning the principal future perspectives on novel interventions on comorbidities in individuals with Fetal Alcohol Spectrum Disorder. The explanations of obtained results was reported in the Discussion Section
Reviewer 3 Report
Comments and Suggestions for Authors
The systematic review entitled "Novel interventions on comorbidities in patients with Fetal Alcohol Spectrum Disorder (FASD): an integrative review" (biomedicines-2841199) by V. Andreu-Fernández et al. is a detailed and up-to-date work in which they review all aspects of Fetal Alcohol Spectrum Disorders (FASD). The review is very well structured, the tables included summarize the available information in an easy-to-read way. It is well written and easy to read and understand.
I believe that the study may be of interest to multiple medical specialties as well as to most of the scientific community.
Comments on the Quality of English LanguageNo comments
Author Response
Reviewer 3: The systematic review entitled "Novel interventions on comorbidities in patients with Fetal Alcohol Spectrum Disorder (FASD): an integrative review" (biomedicines-2841199) by V. Andreu-Fernández et al. is a detailed and up-to-date work in which they review all aspects of Fetal Alcohol Spectrum Disorders (FASD). The review is very well structured, the tables included summarize the available information in an easy-to-read way. It is well written and easy to read and understand. I believe that the study may be of interest to multiple medical specialties as well as to most of the scientific community.
We would like to thank the reviewer for appreciating our study.
Reviewer 4 Report
Comments and Suggestions for Authors
There's no approved pharmacological intervention to address the cellular and tissue damage which triggers FASD. However, recent studies included in this review have performed multidisciplinary approaches involving psycho-social interventions and pharmacological treatments aims to improve patients' quality of life and manage associated symptoms. Moreover, natural molecules such as choline or EGCG have shown promising results in recent clinical studies, so they could be used as a complementary treatment for some of the clinical manifestations in patients with FASD. However, further research and longer-term follow up need to be performed to better understand the efficacy, safety, and long-lasting impact of these interventions in FASD patients. This integrative review serves as a crucial reference for clinicians and researchers seeking insights into the novel approach of pharmacological interventions for FASD and associated comorbidities, driving the need to investigate in more targeted and effective treatments for affected individuals.
Although the review article was well-written, one concern is to expand the view by the authors how to address this problem for further studies.
Author Response
There's no approved pharmacological intervention to address the cellular and tissue damage which triggers FASD. However, recent studies included in this review have performed multidisciplinary approaches involving psycho-social interventions and pharmacological treatments aims to improve patients' quality of life and manage associated symptoms. Moreover, natural molecules such as choline or EGCG have shown promising results in recent clinical studies, so they could be used as a complementary treatment for some of the clinical manifestations in patients with FASD. However, further research and longer-term follow up need to be performed to better understand the efficacy, safety, and long-lasting impact of these interventions in FASD patients. This integrative review serves as a crucial reference for clinicians and researchers seeking insights into the novel approach of pharmacological interventions for FASD and associated comorbidities, driving the need to investigate in more targeted and effective treatments for affected individuals.
- Although the review article was well-written, one concern is to expand the view by the authors how to address this problem for further studies.
The authors, as suggested by the reviewer, expanded the own point of view at lines 410-415
Round 2
Reviewer 2 Report
Comments and Suggestions for Authors
The paper has been improved according to the reviewers' suggestions. It can be accepted in present form.